# Eastern Cape Healthcare Workers Acquisition of SARS-CoV-2 (ECHAS): Cross-Sectional (Nested Cohort) Study Protocol

**DOI:** 10.3390/ijerph18010323

**Published:** 2021-01-05

**Authors:** Oladele Vincent Adeniyi, David Stead, Mandisa Singata-Madliki, Joanne Batting, Leo Hyera, Eloise Jelliman, Shareef Abrahams, Andrew Parrish

**Affiliations:** 1Department of Family Medicine, Cecilia Makiwane Hospital, East London 5219, South Africa; 2Faculty of Health Sciences, Walter Sisulu University, Mthatha 5117, South Africa; dfstead@gmail.com (D.S.); andygp@mweb.co.za (A.P.); 3Department of Internal Medicine, Cecilia Makiwane and Frere Hospitals, East London 5219, South Africa; 4Effective Care Research Unit, Department of Obstetrics & Gynaecology, Frere Hospital, East London 5247, South Africa; mandisa.singata@gmail.com (M.S.-M.); joannebatting@hotmail.com (J.B.); 5Faculty of Health Sciences, University of Fort Hare, East London 5200, South Africa; 6Department of Obstetrics & Gynaecology, Cecilia Makiwane Hospital, East London 5219, South Africa; leohyera@gmail.com; 7Department of Radiology, Frere Hospital, East London 5247, South Africa; jellimaneloise@gmail.com; 8Department of Pathology, Division of Medical Microbiology, National Health Laboratory Service, Port Elizabeth 6020, South Africa; shareef.abrahams@nhls.ac.za

**Keywords:** Eastern Cape, healthcare workers, SARS-CoV-2 RT-PCR, SARS-CoV-2 serology, South Africa

## Abstract

Healthcare workers (HCWs) are at increased risk of infection by the virulent severe acute respiratory syndrome coronavirus-2 (SARS-CoV-2). Though data exist on the positivity rate of the SARS-CoV-2 reverse transcription polymerase chain reaction (RT-PCR) test as well as COVID-19-related deaths amongst HCWs in South Africa, the overall infection rate remains underestimated by these indicators. It is also unclear whether the humoral immune response after SARS-CoV-2 infection offers durable protection against reinfection. This study will assess the SARS-CoV-2 seroprevalence amongst HCWs in the Eastern Cape (EC) and examine the longitudinal changes (rate of decay) in the antibody levels after infection in this cohort. Using a multi-stage cluster sampling of healthcare workers in selected health facilities in the EC, a cross-sectional study of 2250 participants will be recruited. In order to assess the community infection rate, 750 antenatal women in the same settings will be recruited. Relevant demographic and clinical characteristics will be obtained by a self-administered questionnaire. A chemiluminescent microparticle immunoassay (CMIA) will be used for the qualitative detection of IgG antibodies against SARS-CoV-2 nucleocapsid protein. A nested cohort study will be conducted by performing eight-weekly antibody assays (X2) from 201 participants who tested positive for both SARS-CoV-2 RT-PCR and serology. Logistic regression models will be fitted to identify the independent risk factors for SARS-CoV-2 infection. The cumulative SARS-CoV-2 infection rate and infection fatality rate among the frontline HCWs will be estimated. In addition, the study will highlight the overall effectiveness of infection prevention and control measures (IPC) per exposure sites/wards at the selected health facilities. Findings will inform the South African Department of Health’s policies on how to protect HCWs better as the country prepares for the second wave of the SARS-CoV pandemic.

## 1. Introduction

Healthcare workers (HCWs) are a high-risk group for severe acute respiratory syndrome Coronavirus-2 (SARS-CoV-2) infection owing to occupational exposure to large numbers of highly infectious patients with COVID-19 disease (mostly pneumonia) requiring hospitalization for supportive oxygen therapy [1]. Despite infection prevention and control (IPC) measures including isolation of cases and the use of personal protective equipment (PPE), HCWs still acquire the infection at a higher rate than the general population. A prospective study of 200 frontline HCWs in London during the peak of viral transmission showed that 44% became infected, more than double that of the local population [2]. A smartphone application survey of almost 100,000 HCWs from the United Kingdom (UK) and United States of America (USA), and over two million members of the general public, with self-reporting of positive reverse transcription-polymerase chain reaction (RT-PCR) for the SARS-CoV-2 infections was performed [3]. It found an almost twelve-fold increased risk (HR = 11.6, 95% CI: 10.9–12.3) of acquiring SARS-CoV-2 among the frontline healthcare workers (HCWs) in comparison with the general population.

In South Africa, HCWs are considered as priority group for daily symptom screening and testing for any potential COVID-19 symptoms [1]. The cumulative prevalence of SARS-CoV-2 infection among HCWs is unknown, given that PCR test is often limited to symptomatic individuals. While this is a source of concern for administrators of the South African Department of Health, it is not clear whether the hazard risk reported by Nguyen et al. [3] is translatable to HCWs in the sub-Saharan African region, where regional risk may vary. There is value in the local seroprevalence testing of HCWs to assess more adequately the extent of SARS-CoV-2 infections (symptomatic and asymptomatic), and to compare this to a group more representative of the community (ante-natal women for the purpose of this study). Such a study was performed in Denmark, where 28,792 HCWs and 4672 blood donors had SARS-CoV-2 IgM/IgG testing. Seroprevalence was higher in HCWs than in blood donors (4.04% vs. 3.04%; risk ratio (RR) 1.33 (95% CI 1.12–1.58); *p* < 0.001). Male HCWs working in the hospitals, and in dedicated COVID-19 wards, were all significant and independent risk factors for the disease [4].

Due to the high exposure environment of hospitals, accurate data on the cumulative infection rate amongst HCWs will add to the existing literature on SARS-CoV-2 transmission dynamics. Quantifying risk factors for infection amongst HCWs can be instructive for future prevention interventions. Reported inadequate PPE availability and use has been shown to significantly increase risk for infection [3,5]. Within the health facilities, high exposure clinical areas (Accident and Emergency, acute medical ward, and intensive care units) have been associated with increased infections compared to administration or support services [4]. In contrast, other studies have shown no difference between different staff roles, suggesting that most infections are acquired outside of contacts with patients, or outside of the hospital environment [6,7,8].

Having at least one co-morbidity in a HCW was shown to be a significant risk for acquiring COVID-19 in one study [7]. Outside of the health care environment, a general practitioners (GP) network study of 3802 SARS-COV-2 tests performed in the UK, showed significantly increased infections among males, age 40–64 years, black ethnicity, lower socio-economic status, chronic kidney disease patients, and the obese. Interestingly, smokers had a lower risk of infection. [9] (de Lusignan et al. 2020). It is unclear whether the risk factors for acquiring SARS-CoV-2 infection among HCWs in South Africa will follow similar socio-demographic patterns reported in the rest of the world.

Molecular testing (RT-PCR) of respiratory samples (naso/oropharyngeal swab, sputum, and bronchoalveolar lavage) is the recommended diagnostic modality for confirming acute SARS-CoV-2 infection [1]. The limitation of this test, however, is the significant proportion of false negative tests, resulting in under-estimation of true infection prevalence. Asymptomatic infection is also frequent, occurring in 46% (95% CI: 18.48–73.60%) of cases in one meta-analysis [10]. Testing strategies that focus on symptomatic individuals (as were adopted by the South African Department of Health [1]) will always under-estimate the true population burden of the disease and negatively impact on adequate planning of IPC measures for the country.

A wide array of commercial and research SARS-CoV-2 antibody tests have been developed since the identification of the virus. They measure antibodies that target specific viral epitopes (nucleoprotein, spike protein, and receptor-binding domain) and measure immunoglobulin A, M, or G. IgG is the last antibody to rise in response to acute infection, but it persists the longest [11]. Serological testing includes lateral-flow antibody assays, beadbased assays (Luminex technology), enzyme-linked immunosorbent assays (ELISAs), and automated serology platforms. The two basic categories of serological tests currently available for COVID-19 are rapid diagnostic tests (also known as point-of-care tests) and formal laboratory serological tests. Rapid diagnostic tests often use the lateral flow design, which produces a color change on a test strip. The formal assays are either based on the ELISA or the chemiluminescent detection principle. At least three of these assays, available in South Africa, are produced by by Euroimmun, Roche Diagnostics, and Abbott Diagnostics.

A Cochrane review of 54 studies of laboratory and point-of-care assays reported pooled results for IgG/IgM sensitivity of 30.1% (95% CI 21.4–40.7) for 1 to 7 days, 72.2% (95% CI 63.5–79.5) for 8 to 14 days, and 91.4% (95% CI 87.0–94.4) for 15 to 21 days after the onset of illness. Between 21 and 35 days, pooled sensitivities for IgG/IgM were 96.0% (95% CI 90.6–98.3) [11]. Antibody testing is therefore only recommended after 14 days of symptoms in acute infection and has greater utility in late infection or to assess previous SARS-CoV-2 exposure or in seroprevalence surveillance.

Areas of uncertainty with SARS-CoV-2 antibody testing, is how long the IgG antibody levels remain detectable in the blood and the durability of humoral immune response post-infection. Early reports of serial antibody testing in recovering COVID-19 cases indicated an apparently short-lived antibody response. One study of 34 mild cases reported a half-life of 36 days for IgG levels measured over approximately 90 days [12]. In contrast, a recent seroprevalence study was performed on over 30,000 Icelanders, including 1797 RT-PCR confirmed COVID-19 cases, using six different antibody assays. This study demonstrated good IgG antibody durability, with levels rising to two months post-infection, and then sustained up to four months (the study cut-off) [13]. It is therefore unclear whether there are regional or ethnic variations in the durability of SARS-CoV-2 antibodies. The proposed study aims to estimate the cumulative SARS-CoV-2 infection rate and will further examine the durability of the humoral immune response specific to this virus in a cohort of HCWs in the Eastern Cape, South Africa.

## 2. Protocol

This study protocol will be implemented in accordance with the recommendations outlined in the STROBES (Strengthening the Reporting of Observational Studies in Epidemiology) checklist. The use of the STROBES framework will guarantee a high scientific standard.

### 2.1. Design and Settings

This study will adopt a cross-sectional survey with a nested prospective cohort component. Four health facilities have been purposively selected across the Eastern Cape Province, South Africa, for this study. All the three tiers of health care facilities are represented; Frere Hospital (tertiary), Cecilia Makiwane Hospital (regional), and two primary healthcare centers: Nontyatyambo and Empilweni Gompo Community Health Centers (CHCs). Findings will provide guidance on the overall infection rates amongst HCWs across the three tiers of health facilities in the province.

### 2.2. Participants

The study population will include all categories of HCWs at the selected health facilities. Given the large number of HCWs (doctors, nurses, allied health workers, and administrative staff) in the regional (about 1400 staff) and tertiary (about 1500 staff) hospitals in comparison to those in the primary health care (about 120 staff), it is expected that the majority of the participants in this study will be recruited from the two large hospitals. Since this cross-sectional study aims to provide mass screening (using a SARS-CoV-2 serological test) for all HCWs, about 2250 HCWs are expected to take part in this study. The estimated number of participants per study site is proportional to the overall count of HCWs at each site. A total of 750 antenatal women will be recruited from the study sites to provide data on the community infection rate. The recruitment of HCWs is pre-stratified for inclusivity and is proportionate to the total head count of staff in each facility as shown in Table 1.

### 2.3. Study Procedure

All HCWs will be eligible to participate in this study. Using a multi-stage cluster sampling technique, participants’ clusters will be categorized by the facility of employment (Table 1) and exposure areas. Participants will be conveniently sampled within each cluster. Every HCW in each cluster will be given an equal opportunity to be sampled into the study, and participation will be purely voluntary.

Participants will be recruited from the study sites concurrently in order to avoid time variations in the main outcome measures. For inclusivity, exposure areas are pre-defined in accordance with risk assessment by Iversen et al. [4]:High risk: Accident and Emergency unit, acute respiratory (person under investigation/COVID-19) wards and intensive care units (ICU).Intermediate risk: non-respiratory admission wards, outpatient departments (OPDs) and other clinical areas.Low risk: administration offices and other non-clinical areas.

Information about this mass screening will be disseminated through the departmental heads and clinical managers. Each working areas/unit will be allocated specific days to give allowance for those on duty as well as those off-duty to participate with minimal interruptions in patient care. Each participant will complete a self-administered questionnaire. In addition, medical records of deceased HCWs during the pandemic will be reviewed in order to estimate the case fatality rate of HCWs in the study sites.

### 2.4. Study Instrument

The questionnaire for this study was purposively designed to capture data that are relevant to the study objectives, using validated measures from the World Health Organization (WHO) STEPwise tool [14], COVID-19 risk assessment tool [15,16], persistence of symptoms [17] and vaccine hesitancy survey tool [18]. The questionnaire comprises variables on demographics, exposure risks, vulnerability risks, prior SARS-CoV-2 PCR tests and results, COVID-19 symptoms and management, post-COVID-19 persistent symptoms, and perception about the COVID-19 vaccine.

### 2.5. Independent Variables

The selection of the parameters (included in the questionnaire) is based largely on the body of evidence from international literature on factors that influence or increase the risks of acquisition of SARS-CoV-2, severe COVID-19 disease, humoral immune response, and perception about vaccines [3,5,9,15,16,18]. Trained research nurses (without any affiliation to the study sites) will measure the height, weight, and mid-upper arm circumference according to standard protocols and the body mass index will be derived.

### 2.6. Validity and Reliability of the Study Instruments

The construct and criterion validity of the instrument have been established by selecting variables that have been used successfully in measuring the outcome measures of this study in multiple studies [3,5,9,15,16,18]. In addition, the instrument has been piloted with five HCWs at one of the study sites and the feedback from the participants was critically reviewed by the investigators, following which adjustments were made in some of the variables. However, the results of the pilot will not be included in the main study.

### 2.7. Blood Sampling

Following aseptic technique, the research nurses will collect up to 5 mL of venous blood in serum separating tube, which will be transported daily to the National Health Laboratory Service (NHLS) for processing. A maximum of three attempts at drawing venous blood samples from the participants per visit will be allowed. If venesection fails after three attempts, the participant will be re-scheduled for another day in order for a doctor (one of the investigators) to draw the blood sample.

For the sub-study (nested cohort study), about 200 participants will be required to attend a follow up during which another round of blood sampling will take place in accordance with standard protocol.

### 2.8. Oropharyngeal Swabs

Participants who report compatible symptoms (fever, cough, sore throat, loss of smell/taste, and shortness of breath) [1] at the time of the study will be offered oropharyngeal swab test for SARS-CoV-2 PCR. This will be captured separately in the research register as well as in the occupational health and safety (OHS) database for follow-up. Where there are gaps, the NHLS database and the Occupational Health Unit’s records will be reviewed for additional data.

### 2.9. Possible Outcomes

SARS-CoV-2 serology results will be matched with the baseline results of SARS-CoV-2 RT-PCR if previously done.

The following categories will emerge:Both SARS-CoV-2 RT-PCR and SARS-CoV-2 serology are positive: COVID-19 diagnosis confirmed, and humoral immune response is present.SARS-CoV-2 RT-PCR is positive but SARS-CoV-2 serology is negative: COVID -19 diagnosis confirmed but there is no persistent humoral immune response or too early for antibody detection.SARS-CoV-2 RT-PCR not done or negative but SARS-CoV-2 serology is positive: COVID-19 diagnosis was missed by PCR but there is a humoral immune response present.Both SARS-CoV-2 RT-PCR and SARS-CoV-2 serology are negative: no confirmed infection with SARS-CoV-2.

All participants will receive their results within a month of the study from designated persons (Occupational Health Champions) at each study site. Additional interventions (clinical consultations and/or psychologist’s or Employee Assistant Programme practitioner’s referral) will be available for HCWs with specific needs.

### 2.10. Nested Cohort Study (N_b_ = 201 Participants)

The sample size for the cohort study was estimated as 201 based on the expected proportion of HCWs (45% was reported in hospital setting in London by Houlihan et al. [2]) who could have acquired SARS-CoV-2 during the peak period of the pandemic. This sample was calculated at a 95% confidence level and confidence interval of 7.5. The final sample was adjusted in anticipation of a 27% attrition rate during the follow-up of the participants.

Only participants in category one will be pooled into the nested cohort study for serial antibody screening at eight-week interval to monitor the rate of decline in the antibody positivity rate (durability of SARS-CoV-2 specific humoral immune response). The participants in this sub-study will have blood samples drawn on two occasions; at baseline and final sample will be drawn at eight-week interval. Most studies have reported significant decline in the antibody levels by three to four months [12,13]. In anticipation of the challenges of follow-up study, participant recruitment will be based on their willingness to allow monitoring of their SARS-CoV-2 antibody levels. There will be no form of coercion of any HCW to participate or continue follow-up. Participants in this sub-study will receive text messages and face-to-face reminders on the date for blood tests.

### 2.11. Laboratory Testing

Serum samples will be run on the Abbott ARCHITECT *i*1000SR instrument using the Abbott SARS-CoV-2 IgG assay in accordance with the manufacturer’s instructions. This is a CLIA for the qualitative detection of IgG in human serum or plasma against the SARS-CoV-2 nucleoprotein. Strength of response in relative light units reflects quantity of IgG present and is compared to a calibrator to determine the calculated index (specimen/calibrator [S/C]) for a sample (with positive at 1.4 or greater). This assay was independently evaluated for analytic performance and found to have a specificity of 99.9% from 1020 pre-COVID-19 serum specimens and a sensitivity of 100% at 17 days after symptom onset and 13 days after PCR positivity [19].

### 2.12. Main Outcome Measures

Seroprevalence of and factors associated with SARS-CoV-2 infection: overall infection rate and case-fatality rateFactors associated with durable SARS-CoV-2 specific IgG antibodies

## 3. Statistical Analysis

Data will be exported from Redcap to the STATA Version 15 (Stata Corp., College Station, TX, USA) and cross-checked for completeness and accuracy. Incomplete data will be re-verified, and where possible, the OHS and NHLS databases will be reviewed to maximize the chances that data for the main outcome measures are collated accurately. Multi-level analysis will be performed on all the participants; those with complete and incomplete results. Differences in the baseline (demographic and clinical) characteristics of the two groups will be compared by using Pearson chi-square and Fisher’s exact test for bivariate analysis.

### 3.1. Primary Analysis (Point and Cumulative Seroprevalence)

Descriptive statistics (means ± standard deviations for continuous data and counts and proportions for categorical data) will be used to summarize the baseline (demographic and clinical) characteristics of the participants. Cumulative and point seroprevalence indicating the overall infection rate will be estimated. In addition, the case fatality rate per health facility will be estimated based on the total number of deaths of HCWs and the overall infection rate from this study.

The cumulative outcome (positive result obtained by either or both of SARS-CoV-2 RT-PCR and serology) will be described by fitting the univariate logistic model to examine the association with the individual covariates. In addition, multiple logistic regression models will be fitted to identify factors that are independently and significantly associated with the outcome measures. All variables will be included in the logistic regression model, then, variables with missing data will be excluded in the second model. Sensitivity analysis will be performed using complete data only, with an assumption that missing data would have occurred at random.

In addition, the symptom data will be correlated with the composite results of SARS-CoV-2 PCR and serology tests. An epidemiological curve will be constructed by including data of all infected HCWs with SARS-CoV-2 PCR-positive tests.

### 3.2. Sub-Analysis (Longitudinal Changes in Antibody Detection)

Baseline detection of SARS-CoV-2 specific antibodies (IgG) will be summarized using descriptive statistics (counts and proportions for categorical data). HCWs with confirmed diagnosis with SARS-CoV-2 RT-PCR will be evaluated for the presence or absence of antibodies and the time interval between the two tests will be estimated. Durability of SARS-CoV-2 specific IgG antibodies will be assessed first by estimating the frequency (percentage) of the presence of antibodies and the mean (standard deviation) duration of persistence of the antibodies. Presence or absence of SARS-CoV-2 specific antibodies (IgG) (yes/no) at baseline will also be correlated with the disease severity.

Further, the nested cohort study will also provide additional answers regarding the durability of the humoral immune response. Baseline serology results of the 201 participants in the sub-study will be compared with the results of the subsequent serology test done at eight-week interval.

## 4. Ethical Considerations

Ethics approval was obtained from the Walter Sisulu University Faculty of Health Sciences Ethics Committee (Project Identification Code: 087/2020). Permission for implementation of the study has been obtained from the EC Department of Health as well as the clinical governance of the respective study sites. Information about the research will be disseminated through the various heads of departments or area managers. In addition, participants will be expected to sign an informed consent form detailing their voluntary participation in the follow-up study. Participants’ right to privacy and confidentiality of medical information will be respected during and after the study. The names and identification numbers of participants (essential for data linkage of SARS-CoV-2 RT-PCR and serology results) will be captured on paper registers, that link to a unique patient identifying number (PTID). These will be kept securely in the locked research office. Paper based questionnaires will only contain these PTID’s, hence ensuring confidentiality of all clinical and sensitive information. The questionnaire data will be entered onto the REDCap^®^ online database.

The data will be exported weekly by a designated person to STATA Version 15 (Stata Corp., College Station, TX, USA). No unauthorized access will be granted to the study materials. All the soft copies will be password-protected during and after the study. The study will be implemented in accordance with the Helsinki Declaration and Good Clinical Practice governing human research.

## 5. Antibody Results and Compensation Claims

It is important to clarify to the participants that the serology results cannot and should not be submitted for compensation claims from the Department of Labor/Health as proof of COVID-19 disease. However, the results from this study could be used for planning purposes toward improving IPC within the health facilities in the province and the country.

## 6. Study Schedule

This project will be implemented according to the time schedule shown in Table 2.

## 7. Conclusions

The cumulative infection rate among the frontline HCWs will highlight the gaps to be addressed through effective infection prevention and control measures within the selected facilities. In addition, findings of infection rates per exposure areas might guide the personnel allocations within the respective facilities. The South African Department of Health will be assisted with reliable data on the vulnerabilities of its workers to the virus. In addition, the study will elucidate on the durability of humoral specific immunity to SARS-CoV-2 in the African population and may provide further insights for vaccine researches in the African sub-region.

## Figures and Tables

**Table 1 ijerph-18-00323-t001:** Proposed recruitment of healthcare workers (HCWs) into the study.

Sites	HCWs (*n*)	Antenatal Women (*n*)
Frere Hospital	1100	400
Cecilia Makiwane Hospital	1000	350
Nontyatyambo CHC	75	-
Empilweni Gompo CHC	75	-

CHC: Community health centre.

**Table 2 ijerph-18-00323-t002:** Proposed study schedule.

Tasks	Sept., 2020	Oct., 2020	Nov., 2020	Dec., 2020	Jan., 2021	Feb., 2021	Mar., 2021	Apr., 2021
Design of protocol and ethical approvals								
Protocol implementation (baseline recruitment)								
Data analysis								
Follow-up study implementation								
Data analysis of follow-up study								
Stakeholder engagement								
Completion of project								

## Data Availability

There is no data available for this protocol.

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
