# Peer review of "Eastern Cape Healthcare Workers Acquisition of SARS-CoV-2 (ECHAS): Cross-Sectional (Nested Cohort) Study Protocol"

_ijerph, 2021, doi:10.3390/ijerph18010323_

Round 1

Reviewer 1 Report

This is a study protocol detailing a cross sectional study of SARS-CoV-2 infection prevalence and seroprevalence of healthcare workers (HCWs) in South Africa. 

The introduction is well written and highlights the importance of the study. The introduction provides clear contextualisation in the published literature to date on the topic. The methodology is detailed and robust, permitting full reproducibility. The design of the study is clear and appropriate for the aims of the study.

I do not think that significant adjustments to the manuscript are required and that these comments can be addressed at the editing stage:

Page 5, Line 228: I understand that the nested cohort group in the study (n=201) are recruited if they test positive on both SARS-CoV-2 PCR and SARS-CoV-2 IgG. In order to recruit 201 participants, it assumes that almost 1 in 10 HCW will fulfil this criteria. Do the study team have a contingency if numbers fulfilling this category are lower, i.e by widening recruitment to all seropositive HCWs, or expanding the screening cohort in other institutions?

Page 6, Line 271: The methods to estimate effectiveness of infection control prevention and control measures are not fully described. However this does not impact on the main objectives of the study. It might be clarified that this would be exploratory research.

Page 2, line 62 might read: “The incidence and prevalence of infection among HCW is unclear and the resultant impact on service delivery has not been reliably estimated”.  

Author Response

Dear Editor,

        Please find attached the response to reviewers' comments.

Thank you.

Vincent Adeniyi

Reviewer 2 Report

  Congratulations for the project. It is relevant and necessary for the population of that area. The methodology described is adequate and I wish you the best of luck to carry it out. It would be interesting to have a timetable as well as a follow-up plan, but in general the project is wonderful. Congratulations to the team. The only thing that I consider absolutely necessary to change is the title, which must explicitly indicate that it is a protocol to avoid being identified as a study in future bibliographic searches.

Author Response

Dear editor,

      Please see the consolidated response to the reviwers' comments.

Regards

Vincent Adeniyi
